

# On the development of diagnostic support algorithms based on CPET biosignals data *via* machine learning and wavelets

Rafael F. Pinheiro and Rui Fonseca-Pinto

Center for Innovative Care and Health Technology (ciTechCare), School of Health Sciences
(ESSLei), Polytechnic University of Leiria, Leiria, Leiria, Portugal

## ABSTRACT

For preventing health complications and reducing the strain on healthcare systems, early identification of diseases is imperative. In this context, artificial intelligence has become increasingly prominent in the field of medicine, offering essential support for disease diagnosis. This article introduces an algorithm that builds upon an earlier methodology to assess biosignals acquired through cardiopulmonary exercise testing (CPET) for identifying metabolic syndrome (MS), heart failure (HF), and healthy individuals (H). Leveraging support vector machine (SVM) technology, a well-known machine learning classification method, in combination with wavelet transforms for feature extraction, the algorithm takes an innovative approach. The model was trained on CPET data from 45 participants, including 15 with MS, 15 with HF, and 15 healthy controls. For binary classification tasks, the SVM with a polynomial kernel and 5-level wavelet transform (SVM-POL-BW5) outperformed similar methods described in the literature. Moreover, one of the main contributions of this study is the development of a multi-class classification algorithm using the SVM employing a linear kernel and 3-level wavelet transforms (SVM-LIN-MW3), reaching an average accuracy of 95%. In conclusion, the application of SVM-based algorithms combined with wavelet transforms to analyze CPET data shows promise in diagnosing various diseases, highlighting their adaptability and broader potential applications in healthcare.

Corresponding author
Rafael F. Pinheiro,
rafael.f.pinheiro@ipleiria.pt

## INTRODUCTION

Metabolic syndrome (MS) and heart failure (HF) both constitute significant global public health issues. MS is a set[1] of conditions that occur simultaneously and elevate the risk of type 2 diabetes, cardiovascular diseases, and other health problems. According to *Noubiap et al. (2022)*, depending on the criteria applied, the global prevalence of MS can vary from 12.5% to 31.4%. Moreover, prevalence varies across different regions, for example, based on the Adult Treatment Panel (ATP) III definition and the World Health Organization (WHO) regions, the prevalence of MS is 32.9% in the Eastern Mediterranean Region, 26.0% in the Region of the Americas, 25.3% in the European Region, and 18.9% in the African Region. These numbers establish it as one of the most prevalent chronic diseases worldwide, being associated with higher mortality from various causes (*Li et al., 2021*). Conversely, HF is a clinical condition that has a very significant impact on day-to-day life, gradually becoming incapacitating and occurs when the heart is unable to adequately pump blood to supply oxygen and nutrient needs. The prevalence of heart failure is on the rise, affecting approximately 26 million individuals worldwide (*Bowen et al., 2020*). The key to attenuating the societal impact of these conditions lies in prevention, early detection, and appropriate treatment.

Beyond the prevalence issue mentioned above, HF and metabolic syndrome (MS) are chosen for the classification algorithm in this study because of their well-established clinical interconnection. MS is a recognized risk factor for the development of HF and commonly occurs as a comorbidity in HF patients (*Purwowiyoto & Prawara, 2021*), particularly in cases of HF with preserved ejection fraction (HFpEF), where an increased risk of hospitalization is associated with MS (*Zhou et al., 2021*). Although MS can be diagnosed through straightforward clinical measurements, such as waist circumference, fasting glucose, triglycerides, HDL cholesterol, and blood pressure, cardiopulmonary exercise testing (CPET) offers a non-invasive alternative, avoiding the need for invasive blood tests. The use of CPET data in the algorithm aims to improve diagnostic accuracy and inform early intervention strategies, justifying the selection of these two conditions.

CPET is a procedure evaluating the body's reaction to exercise through integrated analysis of cardiovascular, respiratory, and metabolic functions. It furnishes vital insights for diagnosing, prognosticating, and devising treatment strategies for various medical conditions, encompassing cardiac (*Saito et al., 2023*), metabolic (*Rodriguez et al., 2022*), and pulmonary (*Luo et al., 2021*) ailments. Health professionals undertake data interpretation, aiding in evaluating cardiorespiratory capacity, optimizing physical training regimens and diagnosing diseases.

Conversely, comprehensive analysis underlies the interpretation of CPET data, focusing on the variables documented throughout the examination. Presently, interpretation aligns with guidelines and criteria established by medical and exercise physiology associations, as well as by scientific research serving as benchmarks for result comprehension (primarily utilizing the flowchart—refer to *Kaminsky et al. (2017)* and *Hansen et al. (2019)*). The flowchart method uses binary decision trees to classify test results into categories like HF or MS, based on key metrics and normative values (see Fig. 3 in *Brown et al. (2022)*). Along

[1] According to the National Cholesterol Education Program (NCEP), Adult Treatment Panel (ATP) III, individuals are classified as having MS if they meet three or more of the following criteria (*Zhou et al., 2021*): (1) high-density lipoprotein (HDL) cholesterol less than 40 mg/dl (1.02 mmol/l) for men and less than 50 mg/dl (1.29 mmol/l) for women; (2) fasting glucose greater than 100 mg/dl (5.6 mmol/l); (3) triglyceride levels greater than 150 mg/dl (1.7 mmol/l); (4) diastolic blood pressure greater than 85 mmHg or systolic blood pressure greater than 130 mmHg; and (5) abdominal obesity, indicated by a waist circumference greater than 102 cm for men and greater than 88 cm for women.

this vein, it is recognized that CPET data interpretation for diagnostic purposes remains an ongoing discussion, and with advancements in artificial intelligence techniques, novel methods and algorithms have surfaced to assist physicians in delivering more precise diagnoses and treatment plans.

Within the domain of artificial intelligence, classification algorithms have been essential in developing diagnostic tools, particularly using strategies of machine learning and artificial neural networks. In the case of binary classification, the research by *Brown et al. (2022)* in artificial neural networks is noteworthy, wherein hybrid models integrating autoencoders (AE) and convolutional neural networks (CNN) alongside logistic regression (LR) and principal component analysis (PCA) are devised for classifying HF and MS, employing a dataset of 15 CPET files for each condition. Regarding machine learning, in the context of multi-classification, *Inbar et al. (2021)* demonstrates the efficacy of the support vector machine (SVM) technique in categorizing diseases such as heart failure, chronic obstructive pulmonary disease, and healthy volunteers, achieving remarkable accuracy rates of 100% with a training dataset comprising approximately 70 CPET files per ailment. Although the methodologies of these studies differ considerably, both produce highly effective results.

In line with the development of algorithms to support diagnosis using supervised binary classification and multi-classification techniques for HF and MS diseases, this article presents an extended version of *Pinheiro & Fonseca-Pinto (2023)* with significant additional content. *Pinheiro & Fonseca-Pinto (2023)* presented a methodology using 3-level Daubechies wavelet transforms for feature selection and compared the accuracy with the *Brown et al. (2022)* methods, obtaining good results for SVM (HF/MS) binary classification. Now, in this work, the theory advances with new results, for multi-classification considering the HF, MS and health (H) patient labels. The contributions are explained in more detail below:

- Based on the literature review performed by the authors, there are no prior studies that address the creation of algorithms for disease diagnosis using CPET data that integrate SVM with wavelets.
- This work addresses the use of wavelet transforms for preprocessing the data, presenting a method for extracting features from CPET data as a highly efficient alternative for reducing computational costs. An analysis is conducted to compare SVM models with wavelets at three levels and five levels. This method facilitates a considerable reduction in the feature dimensionality employed in classification algorithms for analyzing CPET data. The authors assert that this approach constitutes the primary contribution of the work, offering an efficient algorithm with minimal computational overhead compared to *Inbar et al. (2021)* and *Brown et al. (2022)*.
- This work provides a detailed explanation of the conceptualization of evaluation metrics used (accuracy, precision, recall, F1-score), both for binary and multi-classification cases. During the bibliographic research for this work, a scarcity of content addressing the clear derivation of evaluation metrics for multi-classification was noted. Therefore, the mathematical operationalization for clarification is formalized through Eqs. (1)–(4),

contributing in a didactic context on how to obtain evaluation metrics for multi-classification.

- For binary classification, the SVM model utilizing a polynomial kernel combined with 5-level wavelets (SVM-POL-BW5) presents a new construction methodology and better performance based on the evaluation metrics, in comparison to other algorithms employing LR, CNN, PCA and flowcharts, proving to be able to compete with the AE +LR technique of *Brown et al. (2022)*. For multi-classification, this work with the methodology and application presented (utilization of CPET data) is world-first, with the highest ranked model being the SVM with a linear kernel with wavelets for extracting 3-level features (SVM-LIN-MW3).

- Given that reducing feature dimensions is very important for biosignals originating from the brain (*e.g.*, electroencephalography data), due to the enormous amount of information, it is understood that the methodological approach of SVM with wavelets presented here can be effectively used to support the diagnosis of neurological and psychiatric diseases.

Figure 1 illustrates the design encompassing all the stages of the process, from the collection of CPET data to the post-processing of patient data using SVM with wavelet algorithms. The right side of this figure could be expanded, in the future with new labels, as the authors intend to advance this research by developing a more comprehensive algorithm. This enhanced algorithm will use CPET data to assist in diagnosing a broader range of diseases. The current success of the algorithm in differentiating between MS, HF, and H demonstrates its potential for wider application. The goal is, in the future, to apply this methodology to identify additional conditions, including pulmonary-vascular and mechanical-ventilatory disorders. By incorporating more diverse training data, the algorithm can be further refined to distinguish between a broader spectrum of pathologies and identify patients with several overlapping conditions, thereby enhancing diagnostic accuracy for a great range of diseases.

The rest of the article is structured as follows: the Methods section deals with the main theoretical basis for the development of the algorithms and methodologies are presented; in the Results section, the algorithms developed and their performance are presented with comparisons; the Discussion section presents a brief discussion of some of the issues addressed in this work; and ultimately, the article closes with the Conclusion section, which provides a summary of the findings and future work is proposed.

## METHODS

This section presents all the concepts and methods used to create and validate the algorithms presented in this work.

### Datasets

The dataset selected for this study comes from other datasets from relevant studies carried out previously. Below are details of the original datasets for MS, HF and H, as well as details of the dataset selected for training the algorithms presented in this study.

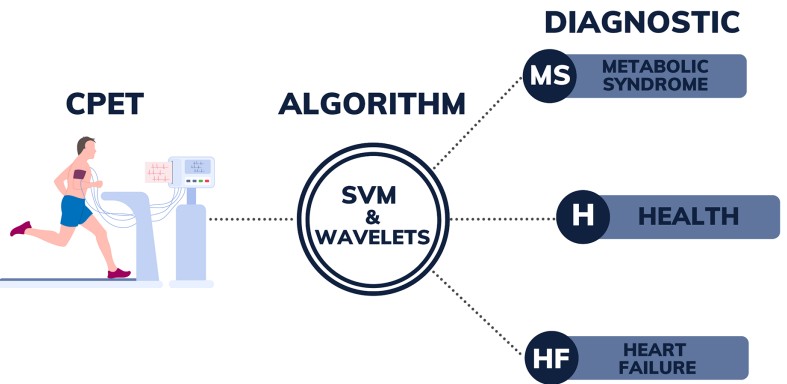

**Figure 1 Illustration of the diagnostic support system from CPET data.**

### Original metabolic syndrome and heart failure datasets

For MS and HF, the data used to develop the algorithms was obtained from rigorously conducted studies supported by renowned institutions, such as the National Institute of Health/National Heart Lung and Blood Institute (NIH/NHLBI) and the American Heart Association. Data for MS was gathered from a research study funded by the (NIH/NHLBI), "Exercise dose and metformin for vascular health in adults with metabolic syndrome". Data on HF originated from patient research funded by the American Heart Association, "Personalized Approach to Cardiac Resynchronization Therapy Using High Dimensional Immunophenotyping", as well as the NIH/NHLBI, "MRI of Mechanical Activation and Scar for Optimal Cardiac Resynchronization Therapy Implementation". For easier access to the data, the article of *Brown et al. (2022)* made the MS and HF dataset available on *GitHub* (https://github.com/suchethassharma/CPET).

The study's eligibility criteria for participants with MS and HF included specific age and health parameters. For MS, adults aged 40 to 70 years with a BMI between 27 and 47 kg/m$^2$ were included, provided they were not diagnosed with Type 2 diabetes and were not engaging in more than 60 min of exercise per week. Participants needed to meet at least three out of five Metabolic Syndrome criteria[1]. Exclusion criteria for MS included morbid obesity, recent significant weight changes, and a history of chronic diseases that could affect study outcomes. For HF, eligible participants were adults over 18 and 85 years old with chronic systolic heart failure and a left ventricular ejection fraction (HFrEF) of 35% or less. Some volunteers also had to meet Class I or IIa directives for the resynchronized cardiac therapy (*Epstein et al., 2008*). HF patients with contraindications to cardiac magnetic resonance imaging (CMR), such as those with implantable devices or certain medical conditions, were excluded. Further details can be found in other articles that were published from the projects mentioned or used their datasets, for example *Gaitán et al. (2019)*, *Heiston et al. (2019)*, *Malin et al. (2019)*, *Bilchick et al. (2020)*, *Gao et al. (2021)*, *Auger et al. (2022)*, *Brown et al. (2022)*, and in the respective projects links provided for MS (https://reporter.nih.gov/search/PoZvzQo230OWPUeVJHEr1g/projectdetails/9934226) and HF (https://reporter.nih.gov/search/u0O4hSNlh0eH8Jj84y-Y0A/projectdetails/9544361).

All patients underwent CPET on a treadmill, following a protocol divided into three phases: rest, exercise, and recovery. During the exercise phase, the treadmill's incline and speed were progressively increased. The Exercise Physiology Laboratory (EPL) at the General Clinical Research Center (GCRC) of the University of Virginia conducted the CPET, ensuring accurate data collection, which includes respiratory measurements and heart rates. All the variables provided by CPET captured from the volunteers with MS and HF, according to *GitHub* (https://github.com/suchethassharma/CPET) are shown in Table 1. Note that this table lists all the variables used in this work, from where they were obtained and the classification application made.

### Original healthy dataset

CPET data from healthy volunteers was obtained from the Exercise Physiology and Human Performance Laboratory at the University of Malaga (UM). The data was found in the Physionet database (*Goldberger et al., 2000*) published by the article (*Mongin, Romero & Cruz, 2021*), as previously used in *Mongin et al. (2021)*. This dataset includes information from 992 exercise tests conducted between 2008 and 2018.

Inclusion criteria required participants to be athletes (amateur or professional), healthy, and aged between 10 and 63 years. Participants had to voluntarily agree to the maximal exercise test, with written informed consent obtained from all participants and legal guardians for those under 18 years of age. There were no specific exclusion criteria mentioned beyond standard safety guidelines for exercise testing. The research adhered to the guidelines of the Declaration of Helsinki and received approval from the Ethics Review Board of the UM, ensuring participant protection and well-being.

The procedure for gathering data required administering a maximal graded exercise test (GET) on a treadmill from the PowerJog J series. Each test began with a warm-up phase, where participants walked at 5 km/h, followed by a continuous or incremental effort, with step increments ranging from 0.5 to 1 km/h. Measurements were recorded continuously, capturing respiratory metrics on a breath-by-breath basis, while heart rate was tracked with a 12-lead ECG apparatus. The test was deemed maximal when the participants' oxygen consumption plateaued, indicating exhaustion. Upon finishing the task, the treadmill's pace was lowered to 5 km/h, and participants continued walking to prevent vasovagal syncope. Measurements were taken using specialized equipment, Including the MedGraphics CPX metabolic gas analysis system and the Mortara ECG machine with 12 leads, all overseen by sports science professionals. Table 1 presents all the variables collected from the CPET of healthy volunteers provided by *Mongin, Romero & Cruz (2021)*.

### Dataset for this work

The dataset for this work, in the treadmill test model, comprised 45 individuals, including 15 with a diagnosis of MS, 15 with HF, and 15 healthy (H) individuals. The dataset for MS and HF originates from the information provided in the subsection "Original Metabolic Syndrome and Heart Failure Datasets" and was obtained from the GitHub repository (https://github.com/suchethassharma/CPET). Meanwhile, the dataset for Healthy (H)

**Table 1 CPET variables considered for this work.**

| Variable description | Abbreviation | Data source | Application |
|---|---|---|---|
| Breath-by-Breath | $Time(min)$ | *GitHub* (https://github.com/suchethassharma/CPET) | Variable not used |
| Time since the measurement starts | $Time(min)$ | *Mongin, Romero & Cruz (2021)* | Variable not used |
| Metabolic equivalents | $METS$ | *GitHub* (https://github.com/suchethassharma/CPET) | Binary classification |
| Heart rate | $HR(beats/min)$ | *GitHub* (https://github.com/suchethassharma/CPET) and *Mongin, Romero & Cruz (2021)* | Binary classification and multi-classification |
| Peak oxygen consumption | $\dot{V}O_2(L/min)$ | *GitHub* (https://github.com/suchethassharma/CPET) and *Mongin, Romero & Cruz (2021)* | Binary classification and multi-classification |
| Peak oxygen consumption is measured in milliliters of oxygen used in 1 min per kilogram of body weight | $\dot{V}O_2/kg((ml/min)/kg)$ | *GitHub* (https://github.com/suchethassharma/CPET) | Variable not used |
| Volume of carbon dioxide released | $\dot{V}CO_2(L/min)$ | *GitHub* (https://github.com/suchethassharma/CPET) and *Mongin, Romero & Cruz (2021)* | Binary classification and multi-classification |
| Respiratory exchange ratio | $RER$ | *GitHub* (https://github.com/suchethassharma/CPET) | Binary classification |
| Ventilation | $VE(L/min)$ | *GitHub* (https://github.com/suchethassharma/CPET) and *Mongin, Romero & Cruz (2021)* | Binary classification and multi-classification |
| Ratio of ventilation by peak oxygen | $VE/\dot{V}O_2$ | *GitHub* (https://github.com/suchethassharma/CPET) | Variable not used |
| Ratio of ventilation by volume of carbon dioxide released | $VE/\dot{V}CO_2$ | *GitHub* (https://github.com/suchethassharma/CPET) | Variable not used |
| Respiratory rate | $RR(breaths/min)$ | *GitHub* (https://github.com/suchethassharma/CPET) and *Mongin, Romero & Cruz (2021)* | Binary classification and multi-classification |
| Expiratory tidal volume (expiratory time) | $Vtex(L)$ | *GitHub* (https://github.com/suchethassharma/CPET) | Binary classification |
| Inspiratory tidal volume (inhale time) | $Vtin(L)$ | *GitHub* (https://github.com/suchethassharma/CPET) | Binary classification and multi-classification |
| Speed of the treadmill (inhale time) | $Speed(mph)$ | *GitHub* (https://github.com/suchethassharma/CPET) | Variable not used |
| Elevation of the treadmill | $Elevation$ | *GitHub* (https://github.com/suchethassharma/CPET) and *Mongin, Romero & Cruz (2021)* | Variable not used |

subjects is based on the details outlined in the subsection "Original Healthy Dataset" and was sourced from the PhysioNet database (*Mongin, Romero & Cruz, 2021*). In the case of H, the 15 oldest volunteers were taken from the original database, in order to have a sample that was closer to MS and HF in terms of age. Thus, the volunteers selected from the original database of healthy people (H) were of ID TEST: 245_3, 296_1, 377_1, 389_1,

**Table 2 Mean and standard deviation of the demographic variables of the volunteers in this study.**

| Condition | HF | MS | H | ALL |
|---|---|---|---|---|
| Sample size | 15 | 15 | 15 | 45 |
| Gender (Female) | 26% | 93% | 26% | 48% |
| Age | 69.3 [58.6, 80.0] | 56.9 [49.9, 63.8] | 55.2 [49.5, 60.9] | 60.5 [50.3, 70.6] |
| Height (cm) | 169.9 [160.2, 179.6] | 155.2 [112.1, 198.2] | 167.3 [159.2, 175.4] | 164.1 [138.0, 190.2] |
| Weight (kg) | 101.8 [81.1, 122.5] | 98.1 [83.2, 113.0] | 67.0 [53.7, 80.3] | 89.0 [66.3, 111.6] |
| BMI | 35.4 [28.2, 42.6] | 35.5 [30.7, 40.2] | 23.7 [20.9, 26.6] | 31.5 [24.0, 39.1] |

390_1, 486_1, 596_1, 597_1, 598_1, 609_1, 651_1, 653_1, 755_1, 756_1, 856_3. Table 2 displays the demographic information of the samples utilized in this research.

Binary classification was performed using HF and MS data, while multi-classification included the dataset of healthy (H) volunteers. The CPET dataset offers a comprehensive array of information garnered during the test. For the binary classification, variables from the CPET were selected as per Table 1. The same table shows the variables used for multi-classification. This selection was guided by the availability of data across both databases. Note that in Table 1, the variables *METS*, *RER*, *Vtex* and *Vtin* were not included in the multi-classification, as they were not provided in the dataset of healthy volunteers.

## Wavelet transforms

Wavelet transforms, a potent mathematical tool extensively employed in signal and image analysis, differ from conventional transforms by offering a multiresolution approach. This approach efficiently captures both local and global signal information. They have emerged as a promising technique in classification algorithms (*Serhal et al., 2022*; *Iniyan, Singh & Hazra, 2023*). Utilizing wavelet technique in data analysis enables the extraction of relevant features across various scales and frequencies, thereby providing a more comprehensive representation of dataset patterns. This capacity to discern discriminative information across multiple resolutions has spurred the creation of more precise and resilient classification models in various domains, including pattern recognition, image processing, and medical diagnostics.

In the application of a wavelet transform, the signal undergoes decomposition into levels ($d_1$, $d_2$, $d_3$,...), each representing details at distinct frequencies. The coefficients within these decompositions elucidate the contributions of each level to the overall portrayal of the original signal. These wavelet coefficients facilitate a detailed examination of the signal across various resolutions. This study employed the Daubechies wavelet of second order with three ($d_1$, $d_2$, $d_3$) and five levels ($d_1$, $d_2$, $d_3$, $d_4$, $d_5$) along with an approximation for the CPET variables ($ap$).

## Features and labels

Features represent the data components utilized in training classification algorithms, while labels denote the classifications assigned to these features. For instance, consider the heart rate (HR) and respiratory rate (RR) data of a patient. Features can encompass the raw

variables themselves (HR and RR), resulting in a sizable dataset. However, for computational efficiency, parameters can be derived from these variables, such as the mean. Consequently, the features for this patient would comprise the mean HR and RR. Conversely, labels correspond to the classifications associated with the patient's features; for instance, a non-diabetic patient may be assigned label 0, whereas a diabetic patient would receive label 1. Hence, a collection of data from multiple patients forms the basis for algorithm training. The larger the patient cohort for training, the more adept the algorithm becomes at disease detection. Further insights into feature extraction are available in *Subasi (2007)*, *Xing et al. (2011)*.

For this work, the features were extracted from the CPET variables presented in Table 1. In order to experiment and identify the best models, the features were organized into two main categories: one set for binary classification tasks and another set for multi-class classification tasks. Figure 2 lists all the types of features used in this work.

In binary classification, three types of features are adopted. The first type, called $X$ (Fig. 2A), is the simplest and consists of the mean and variance of each variable, with the first 15 rows corresponding to the data of HF patients and the other 15 rows for data of MS patients. The $X$ characteristic was constructed using the means and variances of the variables. Figure 2A shows how the data is presented to the algorithm, *i.e.*, an $X$ matrix with a dimension of 30 rows and 16 columns. The second set of features (Fig. 2B), referred to as $BW3$, it includes the average and variance of the wavelet transform coefficients across three levels ($d_1$, $d_2$ and $d_3$), so the matrix of features for this case has a dimension of 30 rows by 64 columns. The last type of feature for binary classification is shown in Fig. 2C, which has five levels and a dimension of 30 rows by 96 columns. For the labels, the first 15 rows correspond to data from HF patients and the other 15 rows to data from MS patients.

In multi-classification, two types of features are extracted. The first type (Fig. 2D), includes the mean and variance of the wavelet transform coefficients of five levels, called $MW5$. The $MW5$ matrix has 45 rows by 60 columns. The second type (Fig. 2E), contains the variance and mean of the wavelet transform coefficients of three levels, called $MW3$. This matrix has 45 rows by 40 columns. For the labels, the first 15 rows correspond to data from patients with HF, from the 16th to the 30th rows correspond to volunteers with MS, and from the 31st to the 45th correspond to healthy volunteers. To improve understanding, the content of the features is summarized with the type of feature, the number of levels and the number of rows in Table 3.

To extract all the features containing wavelets, specific code was used *via* Matlab R2020a that integrated the SVM models to create the diagnostic algorithms that will be presented in the results section.

## Support vector machine

Proposed by *Boser, Guyon & Vapnik (1992)*, the support vector machine (SVM) is a supervised machine learning technique used for both classification and regression, aiming to identify the optimal hyperplane in multidimensional spaces to separate classes of data. Stemming from its effectiveness is the process of enlarging the margin between the support vectors, which represent the nearest points to the decision boundaries (see Fig. 3). This

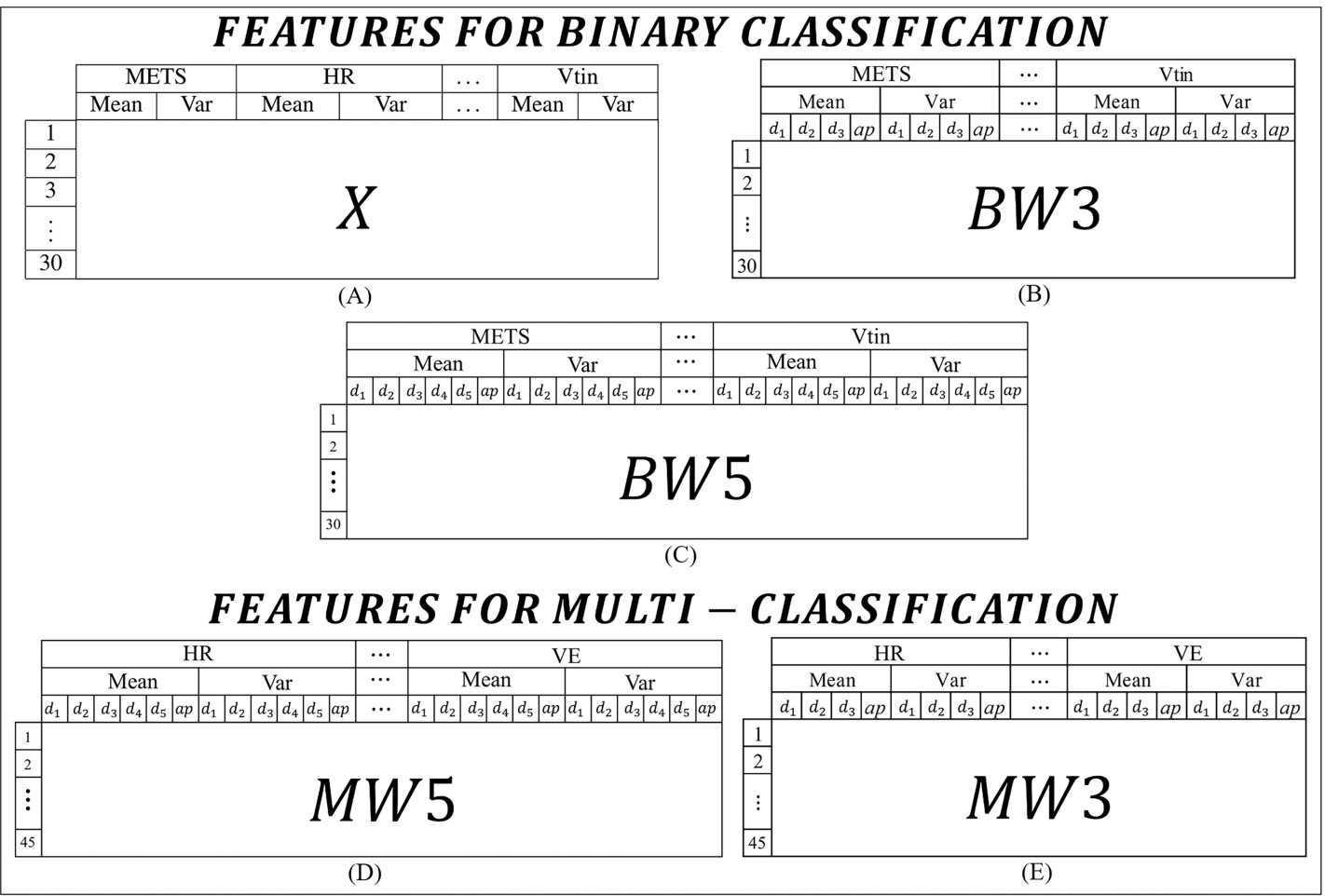

**Figure 2** Features used to build the algorithms: (A) features for binary classification extracted from the means and variances of the CPET variables; (B) and (C) features for binary classification extracted from the means and variances of the coefficients of the wavelet transforms of 3 and 5 levels respectively; (D) and (E) features for binary classification extracted from the means and variances of the coefficients of the wavelet transforms of 5 and 3 levels respectively.

**Table 3** Summary of the features with their applications and dimensions.

| Feature type | Application | Wavelet levels | Rows | Columns |
|---|---|---|---|---|
| X | Binary | Not applicable | 30 | 16 |
| BW3 | Binary | 3 | 30 | 64 |
| BW5 | Binary | 5 | 30 | 96 |
| MW3 | Multi-class | 3 | 45 | 64 |
| MW5 | Multi-class | 5 | 45 | 96 |

approach permits robust generalization even in complex, high-dimensional datasets, rendering it a prominent choice in various data analysis and pattern recognition applications.

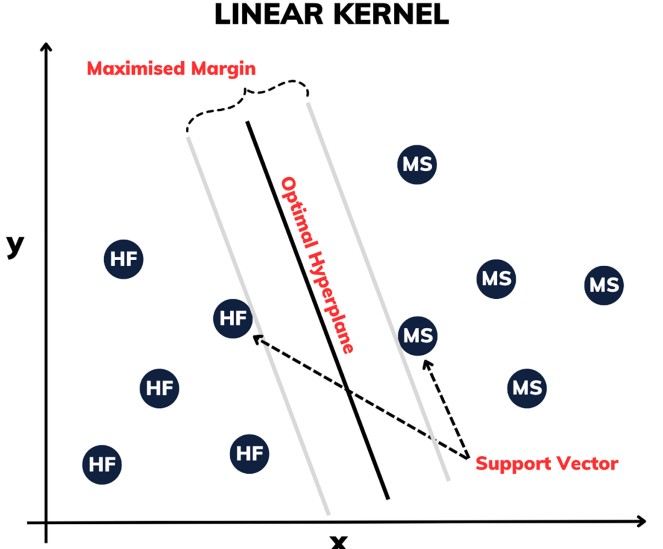

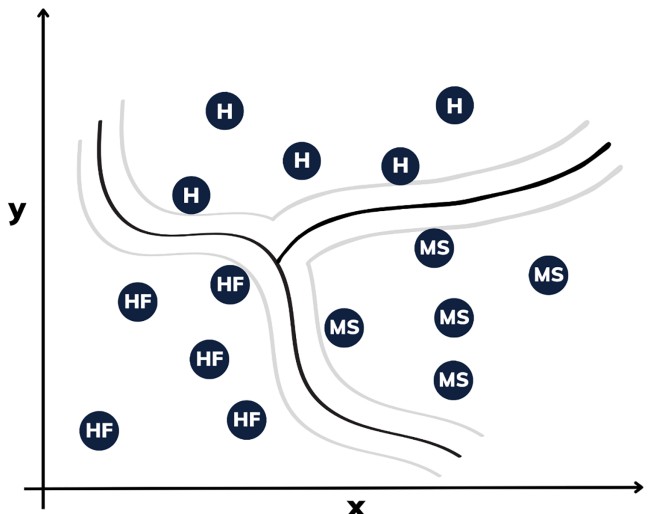

**Figure 3** Illustration of binary classification and multi-classification with different kernels.

Some of the SVM variants are determined by their kernels. In this work, constructions using linear, polynomial and radial basis function (RBF) kernels are made. The linear SVM is used to separate linearly separable classes, while polynomial SVMs are used for non-linearly separable datasets through transformations. The RBF SVM, on the other hand, is highly powerful, mapping the data into a high-dimensional space to separate complex classes.

For multi-classification, several methods have been proposed to combine multiple SVMs from two classes in order to build a multi-class classifier. In this work, using Matlab's *fitcecoc* function, the "one-*vs*-one" with error-correcting output codes (ECOC) model approach is used, which was developed by *Dietterich & Bakiri (1994)* and applied to support vector machines by *Allwein, Schapire & Singer (2000)*.

## Validation process

To validate the algorithm, the cross-validation k-fold was the validation procedure used (see more in *Wong & Yeh (2019)*). The k-fold cross-validation method is a valuable technique in machine learning, especially when working with limited datasets, as is the case in this study. As made by *Pinheiro & Fonseca-Pinto (2023)*, the k-fold cross-validation method splits the dataset into $k$ equal parts. The model is trained $k$ times, using one subset for testing and the remaining $k − 1$ for training. After $k$ iterations, the results are averaged to provide a single performance metric. This ensures efficient data usage and unbiased model evaluation. In this work, a 5-fold cross-validation is applied, where the dataset is divided into five parts. Each model is trained on 4/5 of the data and tested on the remaining 1/5, repeating this process five times. The results are the mean performance of the five rounds of 5-fold cross-validation. For cross-validation, the *crossval* function in

Matlab is used, which does not implement stratification and shuffling automatically (more on this issue is covered in the Discussion section).

The evaluation metrics were gathered following the 5-fold cross-validation process. The metrics used in this work include precision, accuracy, F1-score and recall. These metrics have also been defined and employed in various studies, such as *Brown et al. (2022)* and *Chen et al. (2023)*. The metrics are defined as follows.

- **Accuracy** indicates the model's performance as a whole by calculating the ratio of correct classifications (positive and negative) to the total number of predictions. Although it provides a good general overview, it may be misleading, for example, in cases of class imbalance, descriptors overlapped or outliers (*Morales et al., 2020*; *Michelucci et al., 2021*).
- **Precision** calculates the ratio of true positives to all positive predictions made by the model, indicating the number of predicted positives that are truly positives.
- **Recall**, also known as revocation or sensitivity, measures the percentage of true positive cases correctly detected by the model. This is crucial in scenarios where failing to identify positive cases is critical (False Negatives).
- **F1-score** is the harmonic mean of precision and recall, and offers a single metric that balances the trade-off that exists between them. A low F1-score indicates that either precision or recall is low, making it a useful measure when both metrics are important.

Each formula for these metrics is derived from the confusion matrix (see Fig. 4).

For a binary classification, the confusion matrix generated has a dimension of $2 \times 2$, where the values of the evaluation metrics ($A$, $R$, $P$ and $F1$) are obtained directly according to Fig. 4. In the case of multi-classification, the confusion matrix will be dimensioned according to the number of classes (labels). In order to extract the evaluation metrics in multi-classification, it is necessary to reduce the confusion matrix for each classification label, obtaining the shape of Fig. 4. In this work, to calculate the evaluation metrics in multi-classification ($A^m$, $P^m$, $R^m$ and $F1^m$), one considers obtaining a real overall confusion matrix $C_{3\times3}$, where the values $TP^m$, $FP^m$, $FN^m$ and $TN^m$ are obtained according to Eqs. (1)–(4), respectively, where, if $m = 1$, one has label HF; if $m = 2$, the label is MS; and if $m = 3$, the label is H. For a number of $n$ labels, take $m = 1, 2, 3, \ldots, n$.

$$TP^m = C_{m,m}, \tag{1}$$

$$FP^m = \sum_{j=1}^{n} C_{m,j} - TP^m, \tag{2}$$

$$FN^m = \sum_{i=1}^{n} C_{i,m} - TP^m, \tag{3}$$

$$TN^m = \sum_{i=1}^{n} \sum_{j=1}^{n} C_{i,j} - TP^m - FP^m - FN^m. \tag{4}$$

**Figure 4** Confusion matrix and formulas of the evaluation metrics.

## RESULTS

Figure 5 presents the algorithms developed in this work (double circles in the figure). Moreover, it shows the process of construction and validation, including the main Matlab functions used. The diagram covers the entire structure that encompasses the phases of reading the raw data, creating the features, creating the SVM model and validation. The remainder of this results section is divided into two parts. The first part covers the performance of the algorithms in binary classification, including comparisons with other results in the literature. The second part presents the performance of the algorithms in multi-classification.

### Results for binary classification

Table 4 shows the evaluation metrics of the binary classification algorithms (HF or MS) created in this study. For the model with feature X, the SVM algorithm with a linear kernel (SVM-LIN-X) gave the best result compared to the polynomial kernel and RBF algorithms. Looking at this table, it can be seen that the SVM model with a polynomial kernel for binary classification with 5-level wavelets (SVM-POL-BW5) is the best performer (highlighted in bold). Also note that the confusion matrix generated in this case has dimension $2 \times 2$, so the metrics were calculated by directly applying the formulae in Fig. 4.

Table 5 shows comparisons of the metrics of the SVM-POL-BW5 algorithm (the best performer according to Table 4) with other algorithms developed in other articles. The algorithms being compared are the Flowchart method, convolutional neural networks (CNN), and hybrids: principal component analysis with logistic regression (PCA+LR); and autoencoder with logistic regression (AE+LR). It should be noted that this comparison was conducted honestly, *i.e.*, the same databases were used for all the algorithms, which also carried out the binary classification of HF and MS diseases. It can be seen from this table

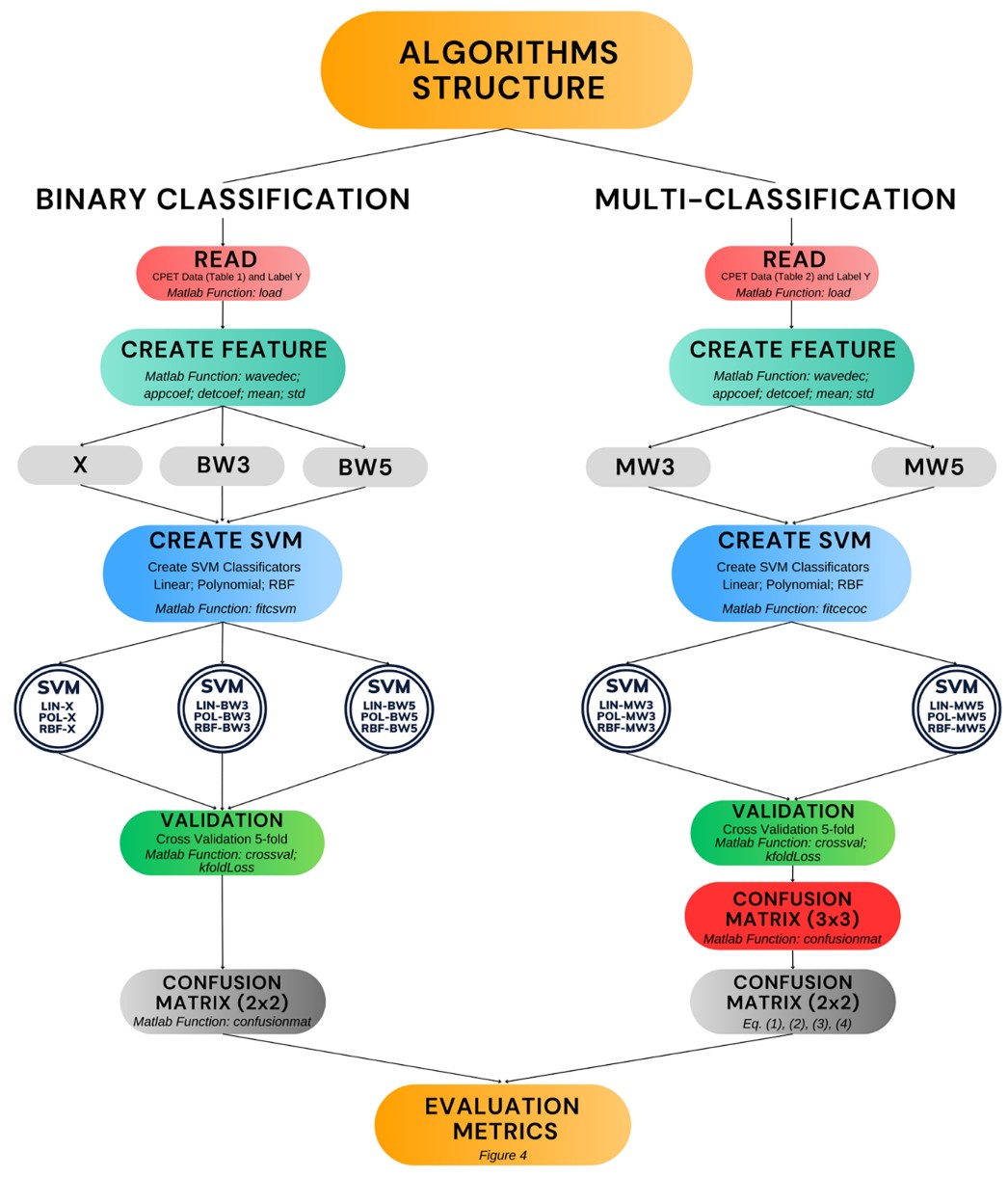

**Figure 5 Algorithm development and validation process.**

that the algorithm developed in this study performed better than the other three types of algorithms compared, coming second only to the AE+LR algorithm.

## Results for multi-classification

Table 6 shows the evaluation metrics obtained for the multi-classification algorithms. These metrics were obtained using the Eqs. (1)–(4), and then the equations in Fig. 4 for each label. Next, Table 7 shows the ranking of the algorithms according to their performance. The values for Table 7 were obtained by calculating the simple mean of the results in Table 6, for example, the accuracy of the SVM-LIN-MW3 algorithm was

**Table 4 Comparisons between the algorithms in this study and the number of wavelet levels for binary classification.** Bold indicates the method with the best performance.

| Method | Accuracy (%) | Precision (%) | Recall (%) | F1-score (%) |
|---|---|---|---|---|
| **SVM-POL-BW**(3/5) | 89/**94** | 87/**90** | 93/**100** | 90/**94** |
| SVM-LIN-BW(3/5) | 92/93 | 88/88 | 98/100 | 93/93 |
| SVM-RBF-BW(3/5) | 87/90 | 85/87 | 89/94 | 87/90 |
| SVM-LIN-X | 83 | 91 | 73 | 81 |
| SVM-POL-X | 74 | 74 | 73 | 73 |
| SVM-RBF-X | 71 | 74 | 68 | 70 |

**Table 5 Comparisons with other methods in the literature for binary classification.** Bold indicates the method with the best performance.

| Method | Accuracy (%) | Precision (%) | Recall (%) | F1-score (%) |
|---|---|---|---|---|
| 1. **AE+LR (*Brown et al., 2022*)** | **97** | **94** | **100** | **97** |
| 2. SVM-POL-BW5 | 94 | 90 | 100 | 94 |
| 3. CNN (*Brown et al., 2022*) | 90 | 100 | 80 | 86 |
| 4. PCA+LR (*Brown et al., 2022*) | 90 | 93 | 87 | 90 |
| 5. Flowchart (*Kaminsky et al., 2017*) | 77 | 78 | 93 | 85 |
| 6. Flowchart (*Hansen et al., 2019*) | 70 | 100 | 53 | 70 |

**Table 6 Comparisons between the SVM (multi-classification) algorithms in this study and the number of wavelet levels.** Bold indicates the method with the best performance.

| Method | Label | Accuracy (%) | Precision (%) | Recall (%) | F1-score (%) |
|---|---|---|---|---|---|
| **SVM-LIN-MW**(3/5) | HF | **95**/91 | **95**/93 | **91**/83 | **93**/87 |
| | MS | **93**/88 | **90**/81 | **90**/84 | **90**/83 |
| | H | **97**/97 | **93**/92 | **100**/100 | **96**/95 |
| SVM-POL-MW(3/5) | HF | 87/87 | 73/67 | 86/93 | 79/78 |
| | MS | 80/78 | 86/94 | 65/61 | 74/74 |
| | H | 93/90 | 81/72 | 98/100 | 88/83 |
| SVM-RBF-MW(3/5) | HF | 46/44 | 81/69 | 36/33 | 49/45 |
| | MS | 64/56 | 24/33 | 34/35 | 28/33 |
| | H | 68/61 | 13/17 | 68/7 | 20/10 |

obtained by calculating the mean of the accuracy of HF, MS and H, *i.e.*, $(95 + 93 + 97)/3$. Finally, Fig. 6 shows a multicriteria analysis chart built from the results of Table 7. In this way, it is possible to see graphically the performance of each algorithm based on its domain region, *i.e.*, the larger the region of coverage, the better the algorithm's performance.

## DISCUSSION

Firstly, this work clarifies some issues that arise in *Pinheiro & Fonseca-Pinto (2023)*. The first concerns the inclusion of data from healthy volunteers; the second is the verification of

**Table 7 The ranking of the multi-classification algorithms for diagnostic support developed in this study.** Bold indicates the method with the best performance.

| Method | Accuracy (%) | Precision (%) | Recall (%) | F1-score (%) |
| --- | --- | --- | --- | --- |
| 1. **SVM-LIN-MW3** | **95** | **93** | **93** | **93** |
| 2. SVM-LIN-MW5 | 92 | 88 | 89 | 88 |
| 3. SVM-POL-MW3 | 86 | 80 | 83 | 80 |
| 4. SVM-POL-MW5 | 85 | 78 | 84 | 78 |
| 5. SVM-RBF-MW3 | 59 | 39 | 46 | 32 |
| 6. SVM-RBF-MW5 | 54 | 40 | 25 | 29 |

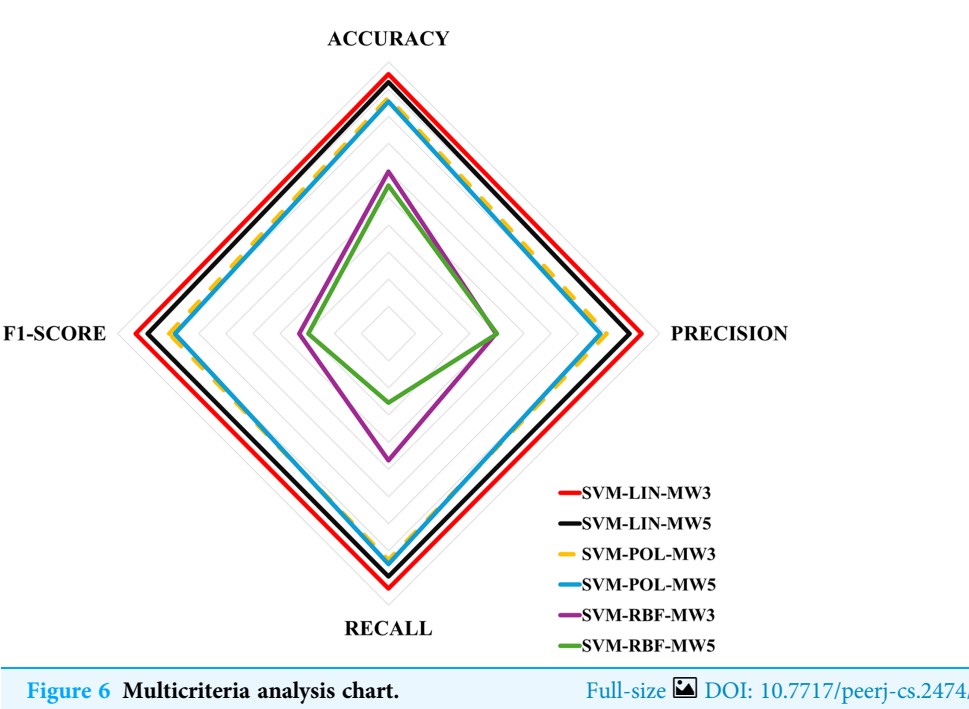

**Figure 6 Multicriteria analysis chart.**

the SVM model with different numbers of wavelet levels; and, finally, the use of more evaluation metrics. In relation to the inclusion of data from healthy volunteers, this work responds by creating a multi-classification algorithm, which is innovative in relation to *Pinheiro & Fonseca-Pinto (2023)* and *Brown et al. (2022)*. With regard to the number of wavelet levels, this work shows that a greater number of wavelet levels does not necessarily improve algorithm performance. For example, in the case of multi-classification, algorithms with three levels of wavelets performed better than algorithms with five levels. Finally, this work provides results with the calculation of more evaluation metrics, in order to confirm the partial results already obtained in *Pinheiro & Fonseca-Pinto (2023)*, and also adds the main metrics for multi-classification.

The results obtained in this research indicate that the proposed methodology, using SVM with wavelet transformations, performed competitively compared to existing solutions in the literature, even on a smaller scale dataset. However, it is important to note

that generalizing this performance for larger datasets requires additional validation. Previous studies, which have applied SVM and wavelets to larger datasets from different domains (*Asgari, Mehrnia & Moussavi, 2015*; *Xin & Zhao, 2017*; *Dhivya & Vasuki, 2018*; *Kehri & Awale, 2018*), suggest that the approach is promising and scalable.

In this study, older volunteers were intentionally selected for the Healthy (H) dataset to obtain a more comparable age range compared to the MS and HF dataset provided by *Brown et al. (2022)*. This strategy was employed to minimize possible age-related bias in the classification results. However, the decision to exclude demographic variables from the feature set is in line with the methodology of *Brown et al. (2022)* and is supported by existing literature, such as the findings of *Inbar et al. (2021)*, which demonstrate that machine learning models can effectively differentiate health states without relying on demographic factors.

In the preparation of data for use in classification algorithms, normalization plays an essential role in ensuring that all variables contribute equally to the decision function of the SVM. This procedure is particularly important when the data exhibit different units of measurement or scales. In this work, normalization was applied using the 'Standardize' option in the *fictsvm* function in Matlab. However, medical standard normalization (such as the 80–100% range of the predicted normal value), as performed by *Inbar et al. (2021)*, was not implemented. This specific type of normalization may further enhance the algorithm's effectiveness and is recommended for future work.

In Table 1, it can be seen that the variables *METS*, *RER*, *Vtex*, and *Vtin* were not included for multi-classification, as they were not provided in the dataset of healthy volunteers (*Mongin, Romero & Cruz, 2021*). However, specifically regarding the *RER* variable, it can be derived from the ratio of $VCO_2$ to $VO_2$. To better understand how the inclusion of *RER* might affect the algorithm's performance, it was incorporated into the model, and simulations for multi-classification with three levels of wavelets were conducted. The results showed that the algorithm's performance was slightly worse. Indeed, the literature supports the notion that variable redundancy, especially through combinations, can degrade algorithm performance and increase the computational cost (*Haq et al., 2019*).

Although this study does not present a detailed analysis of the physiological variables of CPET that are most relevant for a good differentiation between cohorts, the literature reveals some important indications in this regard. Analyzing some studies (*Inbar et al., 2021*; *Portella et al., 2022*; *Zignoli, 2023*), there is an intersection in variables such as $VO_2$, $VCO_2$, $VE$ and $HR$, which are, therefore, of greater importance for classification. According to Table 1, these main variables are considered in the algorithms to extract features for binary classification and multi-class classification.

The algorithm proves useful in real-world scenarios, particularly for patients with HF with preserved ejection fraction who often experience exercise intolerance, a common reason for undergoing CPET. It aids in distinguishing between cardiac and metabolic causes of these symptoms. For MS, the algorithm is a non-invasive approach to early warning of a MS condition, since the standard determination of a MS condition requires clinical analyses with blood sampling focusing on some variables. As various conditions

can influence CPET variables, the algorithm will be further trained to classify additional and overlapping conditions. Future research will focus on expanding the dataset and incorporating data from the CPET of ciTechCare (the authors' institution) installed at Centro Hospitalar de Leiria (Portugal).

When it comes to choosing the best metric for medical diagnosis, sensitivity or recall (R) is highly relevant. False negatives can have serious consequences, for example in the detection of tumors, delaying necessary treatments (*Thölke et al., 2023*; *Spolaôr et al., 2024*). High sensitivity minimizes these errors, allowing for early detection and timely medical intervention, vital for favorable disease outcomes. Although in this study, the task is to identify syndromes early on, whose progression depends on several variables and their stage at the time, it is also essential to analyze them from the point of view of minimizing the number of false negatives.

In this work, the models performed poorly with the RBF kernel, which can be attributed to the linear separability characteristic of the data. The RBF kernel is effective for complex, non-linear patterns, but can be excessively complex and less efficient for data that is (quasi) linearly separable or can be separated with a low-degree polynomial. Furthermore, if the data is dispersed or lacks a clear structure, RBF can generate complicated and inefficient decision boundaries. Under these conditions, simpler kernels, such as linear or polynomial, can offer superior performance in identifying and separating the data (*Gopinath, Kumar & Ramachandran, 2018*; *Kumar, Shukla & Wadhwani, 2024*).

Another interesting issue to discuss is that multi-classification algorithms were developed based on the SVM methodology, despite the fact that, in binary classification, the best SVM model ranked second to the model proposed by *Brown et al. (2022)*, which combines autoencoders with logistic regression. In response to this, it can be argued that the SVM approach was preferred due to its widespread use (already with many paths in Matlab or Python) being an important factor in achieving the main objective of this work, which focuses on multi-classification. AE+LR, on the other hand, leverages artificial neural networks, which come with greater complexity and higher computational costs compared to SVM. Nevertheless, although the SVM approach with wavelets was adopted in this work, future studies could explore the performance of AE+LR as multi-classifiers on CPET data and compare those results with the findings of this study.

Cross-validation in this study was conducted without data shuffling or stratification, which introduces certain limitations. Without shuffling, inherent patterns or ordering within the dataset can skew the results, leading to subsets that may not accurately represent the overall problem, potentially resulting in misleading model evaluations, for example, different accuracies for shuffled and unshuffled (*Chakraborty & Sorwar, 2022*). Additionally, the absence of stratification can lead to imbalanced class distributions across the splits, especially in datasets with class imbalance, making it more challenging for the model to accurately detect minority classes (*Sadaiyandi et al., 2023*). In this work, the data are balanced, which can reduce the inefficiency that may be caused by the lack of stratification. However, for future work, a more rigorous stratification process could be used, for example, using Matlab's *cvpartition* function, which can be used in conjunction with the *crossval* function.

Although there are limitations, the research tackles these issues by replicating the validation process and averaging the results over five rounds of 5-fold cross-validation, so that some random shuffling can take place. In addition, the application of measurements like recall, precision and F1-score provides a more nuanced and extensive evaluation of the model's performance (*Niaz, Shahariar & Patwary, 2022*). This approach offers clearer insights into how effectively the model handles positive cases, false positives, false negatives, and overall class balance, even in the face of data distribution challenges.

A final point to be addressed in this article concerns the limitations of the dataset. The databases used in this study differ, suggesting that a uniform data collection protocol was likely not followed, and the CPET equipment varied across sources. This discrepancy could raise concerns if the method were to be applied directly to real-world scenarios without further validation on larger, more diverse datasets. However, at this stage, it is important to note that this work is best viewed as a pilot study. The investigation is still ongoing, and future efforts should focus on gathering more comprehensive datasets to refine the algorithms and enhance their reliability.

Models trained on small datasets often suffer performance drops due to overfitting and lack of sufficient samples for generalization (*Sordo & Zeng, 2005*; *Prusa, Khoshgoftaar & Seliya, 2015*; *Rahman & Sultana, 2017*). Also, as noted by *Althnian et al. (2021)*, while SVM is relatively robust in such scenarios, relying on support vectors for hyperplane definition, its effectiveness also decreases with reduced data. Fewer samples may result in missing decisive support vectors, impacting the model's performance. Thus, despite being less sensitive than other models, SVM still experiences a decline in smaller datasets. To address this, validating on larger datasets is essential, as limited data may constrain the model's effectiveness. Enhancing generalization and reliability by utilizing larger datasets should be a focus of future research.

## CONCLUSION

This work presented algorithms to support the diagnosis of HF and MS with the option of classifying healthy people. The technique used, SVM with wavelets, proved to be effective for various models, with the SVM-POL-BW5 model excelling for the binary case and the SVM-LIN-MW3 model for the multi-class case. The results presented in this study are promising and motivating to continue this research with the aim of building a more comprehensive model to support the diagnosis of various diseases using CPET data according to Fig. 1.

In this sense, as suggestion for future research, it is proposed to search for more CPET databases with patients diagnosed with HF, MS and other diseases, in order to improve and expand the model proposed in this work. The aim in the future is to create a system capable of integrating existing CPET equipment in health centres, helping doctors to make faster diagnoses and thus improving people's quality of life.

Additionally, a promising direction for future research would be to investigate which CPET parameters aid in creating ideal feature sets to improve model performance. For example, *Schwendinger et al. (2024)* noted the importance of derived variables like oxygen pulse and ventilatory efficiency slopes in machine learning. However, these variables are

derived from other measures, such as oxygen pulse being the ratio of $VO_2$ and $HR$. Therefore, an additional study focused on identifying an optimal set of CPET variables (similarly to what is done directly for features by *Haq et al. (2019)* and *Bezerra et al. (2024)*), combined with the wavelet-based feature extraction technique presented in this work, could lead to an innovative and efficient approach.

Going even further beyond the data provided by CPET, today there is data generated by patients themselves (smartphones, wearables, wristbands, sensor-equipped clothing, among others) that can be shared with healthcare professionals to feed these types of models. In this direction, a new line of research and development can be pursued for the establishment of increasingly personalized and rapid diagnostic support systems.

### Funding
This work was funded by Portuguese national funds provided by the Portuguese Foundation for Science and Technology (FCT) (FCT-UIDB/05704/2020) and in the scope of the research project 2 ARTs (PTDC/EMD-EMD/6588/2020). Rafael F. Pinheiro was supported by FCT through the Institutional Scientific Employment Stimulus CEECINST/00060/2021. The funders had no role in study design, data collection and analysis, decision to publish, or preparation of the manuscript.

### Grant Disclosures
The following grant information was disclosed by the authors:
Portuguese Foundation for Science and Technology (FCT): FCT-UIDB/05704/2020.
Acessing Autonomic Control in Cardiac Rehabilitation (2 ARTS): PTDC/EMD-EMD/6588/2020.
FCT through the Institutional Scientific Employment Stimulus: CEECINST/00060/2021.

### Competing Interests
The authors declare that they have no competing interests.

### Author Contributions
- Rafael F. Pinheiro conceived and designed the experiments, performed the experiments, analyzed the data, performed the computation work, prepared figures and/or tables, authored or reviewed drafts of the article, and approved the final draft.
- Rui Fonseca-Pinto conceived and designed the experiments, analyzed the data, authored or reviewed drafts of the article, and approved the final draft.

### Data Availability
The CPET raw data are available at GitHub: https://github.com/suchethassharma/CPET.

The CPET data for Healthy (H) volunteers is available at Physionet: https://physionet.org/content/treadmill-exercise-cardioresp/1.0.1.

The algorithm codes are available in the Supplemental Files.

## Supplemental Information

Supplemental information for this article can be found online at http://dx.doi.org/10.7717/peerj-cs.2474#supplemental-information.

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
