# Peer review of "On the development of diagnostic support algorithms based on CPET biosignals data via machine learning and wavelets"

_PeerJ Computer Science, doi:10.7717/peerj-cs.2474_

## Round 0.1 · original submission · Major Revisions

The article addresses an interesting topic but requires significant improvements before it can be considered for publication. Specifically, to allow for a better assessment from the reviewers, the authors need to substantially expand the dataset description and provide a detailed explanation of their experimental workflow. Additionally, they should explicitly highlight the limitations of their study. The motivation behind the work should be clarified, and its applicability to real-world settings needs to be thoroughly discussed.

Reviewer 1 ·

Basic reporting

Starting from previous works, this manuscript introduces a new methodology for the classification of metabolic syndrome (MS), heart failure (HF), and healthy people (H), using CPET data. The proposed methodology involves the combination of a well-known supervised machine learning technique, Support Vector Machines (SVMs), already widely used in the medical field, and wavelet transformations, a technique for feature extraction. The promising results achieved are compared with other similar works and showcase the effectiveness of the proposed method.
The article is well-written and provides sufficient context to understand the problem.
Some statements and claims could benefit from additional references to further support the arguments.
The abstract and the introduction could be rephrased since they are quite similar to the previous work of the authors.
The explanation of the proposed methodology is straightforward, well-presented, and sufficiently comprehensive to allow the reader to grasp the main concepts of the techniques adopted.
Furthermore, the authors also provided the code and the dataset to reproduce the experiments, aside from a codebook to better understand the dataset. I did not test the code, but I noticed that it is well-commented, making it easy for users to understand and replicate the experiments.
Overall, the findings of this work make it of interest to this journal.

Experimental design

The experimental design is well-structured and appropriately addresses the research question. However, there are some parts that need further clarification.
1. The descriptions of the metrics are not quite clear, in some cases there are repeated tenses. Maybe they require a little rephrasing.
2. In the description of the accuracy, it is stated that “there may be situations in which it is misleading”. I think this could be further elaborated and supported by some references.
3. The procedure adopted for the experiments is widely used and the use of cross-validation is the correct choice, given the small amount of data available. However, it is not mentioned whether it is implemented with stratified sampling to ensure an equal distribution of the classes, and also if the dataset is shuffled before being divided.
4. In the results for the binary classification, it is said that only the result of the SVM algorithm with a linear kernel is shown for the model with feature X. However, I think that it could be interesting to see also the other models’ performances.
5. Table 4 shows that AE + LR outperforms the approach described in this article, despite the small dataset used. This seems to conflict with what is mentioned in the Discussion section, so it should be further elaborated.
6. It is stated that SVM-POL-BW5 is the best model, but it’s not clear based on what (accuracy? f1?). Which metric is most important in this study? This should be explicated.
7. In Table 5, it strikes me that SVM-LIN-MW (3/5) obtained a recall of 1/1%, as well as SVM-LIN-MW 5. If this is not an error, it should be discussed. A few words could be spent also on the poor performances of SVM-RBF-MW (3/5).

Validity of the findings

The experimental campaign supports the authors’ claim about the competitiveness of their methodology, and the results are competitive with existing solutions in the literature and are relevant to the community. However, the claim in the discussion section might be too strong since the methodology was tested only on a small dataset, and there is no proof that it will perform well also on larger datasets. This claim should be supported by citing works that applied SVM and wavelet transformations to larger datasets (not necessarily medical ones), for example: Xin, Y. and Zhao, Y.’s “Paroxysmal atrial fibrillation recognition based on multi-scale wavelet α-entropy” (2017), or Çomak E. et al’s “Automatic detection of atrial fibrillation using stationary wavelet transform and support vector machine” (2015).

Aside from that, a great limitation of this study is the small amount of data employed in the study. The authors mentioned this briefly at the end of the Discussion section, however, I believe it deserves a further explanation of why it reduces the effectiveness of the models, possibly with some examples and/or references.

Additional comments

Here are some detailed comments:
1. There are some typos/misspelled words:
- line 21 (abstract): … than "others" methodologies…
- line 23 (abstract): … achieved "an mean" accuracy
- line 91: … ("ulitilisation" on CPET data) …
- lines 200-202: inconsistent use of verb tenses
- line 236: … Moreover, shows the … [missing subject]
- line 265: … a multicriteria analysis "chat" built …
- several occurrences of "sentitivity" scattered throughout the manuscript
- lines 144-145,147: maybe an inconsistent use of the acronym HH for heart rate, previously (in Table 1) declared as HR
2. lines 153-155: “... various types of features were obtained separated into two large sets …” this sentence is not quite clear to me.
3. Despite the caption, Figure 1 could be a bit confusing, especially the question marks, which may lead the reader to think that there are unknown goals, rather than future goals
4. Figure 2 could be transformed into a table with the type of feature, number of levels, and number of rows, to improve understandability
5. The caption of Figure 4 does not mention the confusion matrix.
6. Lines 221-222: “In the case of multi-classification, the confusion matrix will be dimensioned according to the number of classifications.” I think it should be the number of classes
7. I find Remark 1 a bit misleading and confusing. I suggest removing it.
8. Line 253: since it is used for comparison, a brief explanation of the Flowchart method could be added
9. In Tables 3, 4, and 5 the best result for each column could be highlighted to improve readability

·

Basic reporting

The study by Fernandes Pinheiro and Fonseca-Pinto proposes binary and multi-classification algorithms based on CPET data to identify individuals with metabolic syndrome, heart failure, or those in good health. Although the approach, especially due to its lower computational cost, may have merit when applied to a larger and more heterogeneous population of patients and healthy individuals, I have reservations about the clinical relevance of differentiating between the three investigated conditions and the transfer to real-world settings.

A minor spell check may be required. Raw data is shared. No further comments here at this review stage.

Experimental design

Comment 1:
The authors provide very little information about the datasets used to develop their algorithms. For the reader to get a better understanding of the patients and healthy individuals, I suggest including relevant summary statistics such as demographics, disease-specific parameters, key parameters to judge the data quality of CPET, etc. Moreover, please describe how the authors ensured the data are valid and of sufficient quality, what devices were used to measure gas exchange in healthy individuals and patients, what protocols were used, and what test modality was chosen (i.e., treadmill or cycle ergometer).


Comment 2: Were there any inclusion and exclusion criteria for the data, and did the authors perform any matching based on factors such as age, sex, and body mass index? If these basic parameters were not considered in the analyses, they could potentially explain the classification ability of the algorithms. Please clarify the inclusion and exclusion criteria used for the datasets and whether any matching was done based on age, sex, body mass index, or other relevant factors. Additionally, discuss how these parameters were accounted for in the analyses to ensure the robustness and reliability of the classification algorithms.

Validity of the findings

Comment 3: The selection of metabolic syndrome and heart failure as the conditions for differentiation seems arbitrary. A stronger rationale beyond their prevalence would strengthen the use case for these novel algorithms. Additionally, the practicality of using CPET to diagnose metabolic syndrome is questionable, given that its diagnostic criteria (waist circumference, fasting glucose, triglycerides, HDL, and blood pressure) are relatively simple to measure. Please provide a more detailed rationale for selecting metabolic syndrome and heart failure as the conditions for your algorithms. Discuss the specific advantages and potential clinical significance of using CPET data for these diagnoses. Furthermore, address the usefulness of an algorithm that can differentiate between such heterogeneous conditions, particularly in the presence of other conditions of pulmonary-vascular or mechanical-ventilatory nature. This discussion will help contextualise the real-world applicability and relevance of the proposed algorithms.

Comment 4: The discussion section does not address the physiological aspects of using CPET data and specific features that may be useful for differentiating between the cohorts. Additionally, there is no discussion on the applicability and usefulness of the developed algorithms in real-world settings, where numerous conditions may affect CPET parameters in different ways. Please revise the discussion section to include a paragraph on the physiological basis for the features used in the presented algorithms, especially those that seem particularly relevant for differentiating between the cohorts. Furthermore, discuss the real-world applicability and usefulness of the algorithms, considering the presence of various conditions that could influence CPET parameters differently. This will provide a more comprehensive understanding of the study's implications and potential limitations.

Additional comments

Minor comments:
L. 29: Please accurately define metabolic syndrome.
L. 31: Is there more recent data on the prevalence of metabolic syndrome? Also, prevalence probably varies according to country.

L. 112: Since references are presented in alphabetical order and without numbers, there is no reference linked to ‘1’. Please revise.

L. 116-122: What was the rationale for including only 15 participants of each dataset?

The rationale for selecting specific CPET parameters in Tables 1 and 2 is unclear. For instance, why was the Respiratory Exchange Ratio (RER) included only in the binary classification and not in the multi-classification? Additionally, other parameters such as oxygen pulse or ventilatory efficiency slopes, which have been shown to be valuable for differentiating between various organ system limitations (see, e.g., PMID: 37703323), were not included.

---

## Round 0.2 · Minor Revisions

The authors have made efforts to improve the quality of the article; however, a few minor issues raised by the reviewers still need to be addressed.

Reviewer 1 ·

Basic reporting

The authors have accurately addressed all the comments of the first round of review.
The abstract and the introduction have been rephrased as not to look too similar to the previous work on which this one is based.
Furthermore, they exapanded the reference list with useful citations that support their claims.

Experimental design

The authors improved the issues emerged in the first round of review.
In particular, they clarified their choices and the differences in the performances of the various models.
They also made more visually clear which one perfomed better in the tables.

Validity of the findings

The authors added more references to support their claims in the discussion.
They also expanded the discussion about their work limitations, especially regarding the small amount of data.

Additional comments

I have some final minor comments:
- In the Introduction, "ATP III" is defined in the footnote but not in the main text.
- In the Introduction, the "CPET" acronym is not explained the first time it is used (aside from the abstract)
- If kept, the caption of Figure 2 could be expanded a bit more.
- Since Table 1 and Table 2 have many rows in common, they could be merged into one single table with an additional column including MS/HF/H (or something similar), depending on the availability of the variable. This could also help the reader understand the availability of the data for each subgroup. Another column could be used to indicate in which setting the variable was used (binary classification, multi-classification, none), so that also Table 4 and 5 could be merged into this one. In this way, the reader has all the necessary information about the feature in one single table.
- Regarding stratification, I think MATLAB's cvpartition can be used with crossval, to create a stratified random partition by specifying 'Stratify' as true.

·

Basic reporting

The authors have substantially improved the quality of the manuscript and extensively addressed my previous comments. The manuscript would still benefit from a spell-check.

Experimental design

no comment

Validity of the findings

no comment

Additional comments

no comment

---

## Round 0.3 · accepted · Accept

The authors have answered all reviewers' comments. The manuscript can now be accepted.

Reviewer 1 ·

Basic reporting

The authors have addressed all of my previous comments.
I believe the manuscript is ready to be published.

Experimental design

No further comments.

Validity of the findings

No further comments.

Additional comments

No further comments.